

# The rotor as a sensor — Observing shear and veer from the operational data of a large wind turbine

Marta Bertelè[1,2], Paul J. Meyer[3], Carlo R. Sucameli[1], Johannes Fricke[3], Anna Wegner[3], Julia Gottschall[3], and Carlo L. Bottasso[1]

[1]Wind Energy Institute, Technische Universität München, Boltzmannstr. 15, 85748 Garching bei München, Germany
[2]Siemens Gamesa Renewable Energy, Beim Strohhause 17-31, 20097 Hamburg, Germany
[3]Fraunhofer Institute for Wind Energy Systems IWES, Am Seedeich 45, 27572 Bremerhaven, Germany

**Correspondence:** C. L. Bottasso (carlo.bottasso@tum.de)

**Abstract.**

This paper demonstrates the observation of wind shear and veer directly from the operational response of a wind turbine equipped with blade load sensors. Two independent neural-based observers, one for shear and one for veer, are first trained using a machine learning approach, and then used to produce estimates of these two wind characteristics from measured blade
load harmonics. The study is based on a data set collected at an experimental test site, featuring a highly-instrumented 8 MW wind turbine, an IEC-compliant met mast, and a vertical profiling lidar reaching above the rotor top.

The present study reports the first demonstration of the measurement of wind veer with this technology, and the first validation of shear and veer with respect to lidar measurements spanning the whole rotor height. Results are presented in terms of correlations, exemplary time histories and aggregated statistical metrics. Measurements of shear and veer produced by the
observers are very similar to the ones obtained with the widely adopted profiling lidar, while avoiding its complexity and associated costs.

## 1 Introduction

The goal of this paper is to demonstrate the observation of wind shear and veer, directly from the operational response of a wind turbine. This is achieved by the concept of the rotor as a sensor where the blades, scanning the flow field, are able to measure
relevant characteristics of the inflow. A key advantage of this approach, termed *wind sensing*, is that it does not require extra hardware, but simply relies on the standard operational data available in the supervisory control and data acquisition (SCADA) system, in addition to blade load measurements. Although the latter are not always available on all production turbines, they are becoming more and more widespread, as they are used for other functions, such as load mitigation and condition monitoring.

A novelty of this paper is the first ever – to the authors' knowledge – demonstration of the observation of veer using this
technology. This is made possible by the formulation recently proposed in Kim et al. (2023), where feed-forward neural networks are trained to estimate various wind characteristics from blade load harmonics. This machine-learning approach improves on various previous formulations all based on the use of load harmonics, starting with the study of Bottasso and Riboldi (2014) and then further developed over the years, as more completely described in Bertelè et al. (2021) and references





therein. In addition to vertical shear and veer, load harmonics can be used to estimate horizontal shear and directions (lateral misalignment and upflow). However, load harmonics are not the only way to estimate wind inflow characteristics. For example, Bottasso et al. (2018) use the rotor blades as local scanning wind sensors, which produce estimates of the vertical and horizontal shears, the latter serving also as a wake detector (Schreiber et al., 2020).

A second novelty of the paper is the demonstration of the observation of shear (and veer) over the entire rotor disk. Although the field validation of shear has been reported before, previous studies were based on measurements from met masts reaching only up to hub height (Bertelè et al., 2021; Schreiber et al., 2020), and not to the top of the rotor. The present work is based on the BHV test site (Meyer and Gottschall, 2022), which features the large 8 MW AD8-180 wind turbine, equipped with various sensors that include strain gauges along the blades. The site is complemented by an IEC-compliant met mast and by a vertical-profiling pulsed scanning lidar, capable of reaching well above the rotor top.

Results reported in this paper indicate that both shear and veer can be measured by wind sensing, in general exhibiting a very good match with the corresponding measurements provided by the vertical profiling (VP) lidar, an IEC-approved and widely-adopted device for resource assessment and power performance testing (IEC, 2022). This is remarkable, because – when load sensors are already installed on a turbine – the measurement of these inflow quantities comes at no additional hardware purchase or maintenance cost, as the technology simply amounts to an on-board software upgrade.

Most turbines today operate based only on a very limited knowledge of the ambient conditions, as provided by the on-board anemometry system. Although this is the current standard, it is reasonable to expect that a more complete knowledge of the inflow might improve the way future turbines will be operated. The recent study of Sucameli et al. (2023) presents a good example of why and how a shear and veer observer might be useful. In fact, the authors considered an extensive data set collected during a recent campaign at a site characterized by frequent significant shear and veer. Their analysis indicates that these inflow characteristics can produce lateral wake displacements of the same order of magnitude of those generated by the typical intentional misalignments used in wake-steering control. In such situations, it is clear that neglecting the impact of shear and veer on wake behavior might negatively influence the performance of a wind farm controller, because the effect of a disturbance (shear and veer) is similar to the effect of the control action. The shear and veer observers demonstrated here could provide such information at the rotor disk of each turbine in a park (in contrast to a met mast, which will never be exactly co-located with a turbine, and only rarely will be exactly in front of it), at essentially no cost and with no extra hardware (in contrast with lidar-based solutions).

The paper is organized in two main parts. First, Sect. 2 presents the methods. Section 2.1 reviews the formulation of the shear and veer observers, following the approach developed by Kim et al. (2023). Next, Sect. 2.2 describes the BHV test site and its instrumentation. Finally, Sect. 2.3 describes the calculation of shear and veer from the lidar and mast measurements. This first methodological section is followed by Sect. 3, which presents the results. First, Sect. 3.1 discusses the training of the neural-based observers on a portion of the data set. Next, Sect. 3.2 provides an analysis of their performance on an independent validation data set, considering correlations between estimates and lidar-provided references, an exemplary time history, as well as aggregated statistical quality metrics. Finally, Section 4 concludes the work discussing its main findings.



## 2  Methods

### 2.1  Formulation of the shear and veer observers

Following Kim et al. (2023), the wind observer is formulated as

$$y_{\mathrm{E}} = \mathrm{NN}(\boldsymbol{p}, \boldsymbol{x}_{\mathrm{M}}), \tag{1}$$

where $y$ represents a scalar wind characteristic, $\mathrm{NN}(\cdot, \cdot)$ is a single-output neural network (Bishop, 2006), $\boldsymbol{x}$ is the vector of $N_x$ network inputs, while $(\cdot)_{\mathrm{E}}$ and $(\cdot)_{\mathrm{M}}$ respectively indicate estimated and measured quantities. In this work, two separate networks are considered, one for vertical wind shear $\kappa_v$, and one for vertical wind veer $\Delta\theta^1$.

Considering a single-hidden-layer feed-forward neural network with $M$ hidden neurons, function $\mathrm{NN}(\cdot, \cdot)$ writes

$$\mathrm{NN}(\boldsymbol{p}, \boldsymbol{x}) = \boldsymbol{w}^T \boldsymbol{\sigma}(\mathbf{V}^T \boldsymbol{x} + \boldsymbol{a}) + b, \tag{2}$$

where $\boldsymbol{\sigma}(\cdot)$ is a vector of sigmoid activation functions, $V_{ij}$ and $a_i$ are the synaptic weights and biases connecting the input layer with the hidden layer, whereas $w_i$ and $b$ connect the hidden layer with the output scalar $y$, with $i = [1, M]$ and $j = [1, N_x]$. These free model parameters are stored in vector $\boldsymbol{p} = \{\ldots, w_j, \ldots, V_{ij}, \ldots, a_i, \ldots, b\}^T$.

The network parameters $\boldsymbol{p}$ are trained by backpropagation to minimize the error cost function

$$E(\boldsymbol{p}) = \frac{1}{N} \sum_{l=1}^{N} (y_{\mathrm{E}_l}(\boldsymbol{p}, \boldsymbol{x}_{\mathrm{M}_l}) - y_{\mathrm{M}_l})^2 + W \frac{1}{N_p} \sum_{m=1}^{N_p} p_m^2, \tag{3}$$

where $N_p = M(N_x + 2) + 1$. The first term of the objective function drives the estimates $y_{\mathrm{E}_l}$ produced by the network towards the $N$ available measurements $y_{\mathrm{M}_l}$. The second term of the objective is a Bayesan regularization, which reduces the chances of being trapped in local minima (Burden and Winkler, 2009). The tunable coefficient $W$ sets the relative weight of the Bayesan and error terms. The unknown network weights are iteratively corrected as $\Delta\boldsymbol{p} = -\eta \partial E(\boldsymbol{p})/\partial\boldsymbol{p}$, where $\eta$ is the learning rate. The implementation of the neural network and its training in this paper is based on Matlab (2023).

The network input vector is defined as

$$\boldsymbol{x} = \{\boldsymbol{m}^T, V, \rho\}^T, \tag{4}$$

where $\boldsymbol{m}$ is a vector of blade load harmonics, $V$ is wind speed, and $\rho$ is air density.

Harmonics are computed for the out- and in-plane load components, respectively noted $(\cdot)^{\mathrm{OP}}$ and $(\cdot)^{\mathrm{IP}}$, from the corresponding strain gauge signals via the Coleman-Feingold transformation (Coleman and Feingold, 1958), and then filtered to remove any remaining spurious noise. Following the analysis developed in Kim et al. (2023), only once-per-revolution (1P)

---

[1]As shown in Kim et al. (2023), a similar formulation can be used to estimate the horizontal shear as well as the yaw misalignment angle, although these quantities are not considered further in the present work. In fact, horizontal shear is presumably very modest at the test turbine used here, since it is never waked by other machines. Additionally, only modest variations in yaw misalignment were observed during the present field trials, and therefore the data set does not contain significant-enough information to allow for the identification of a misalignment observer.





harmonics are used for the vertical shear case, and vector $\boldsymbol{m}$ is defined as

$$\boldsymbol{m} = \left\{ m_{1c}^{\mathrm{OP}}, \, m_{1s}^{\mathrm{OP}}, \, m_{1c}^{\mathrm{IP}}, \, m_{1s}^{\mathrm{IP}} \right\}^{T}. \tag{5}$$

On the other hand, the estimation of veer requires a richer input including also the twice-per-revolution (2P) harmonics, leading to the following definition of vector $\boldsymbol{m}$:

$$\boldsymbol{m} = \left\{ m_{1c}^{\mathrm{OP}}, \, m_{1s}^{\mathrm{OP}}, \, m_{1c}^{\mathrm{IP}}, \, m_{1s}^{\mathrm{IP}}, m_{2c}^{\mathrm{OP}}, \, m_{2s}^{\mathrm{OP}}, \, m_{2c}^{\mathrm{IP}}, \, m_{2s}^{\mathrm{IP}} \right\}^{T}. \tag{6}$$

In the previous expressions, subscripts $(\cdot)_{ks}$ and $(\cdot)_{kc}$ respectively indicate kP sine and cosine harmonic amplitudes. A simple explanation of the harmonic content of the shear and veer observers is offered in Appendix A.

The network input vector $\boldsymbol{x}$ of Eq. (4) includes the wind speed $V$ to account for the different behavior, control and deformation of the wind turbine in different operating conditions. This scheduling wind speed is computed as a 30 sec moving average of the rotor-effective wind speed (Soltani et al., 2013). The dependency on air density $\rho$ accounts for the aerodynamic origin of the loads; further details are available in Kim et al. (2023).

A graphical depiction of the neural observers is reported in Fig. 1.

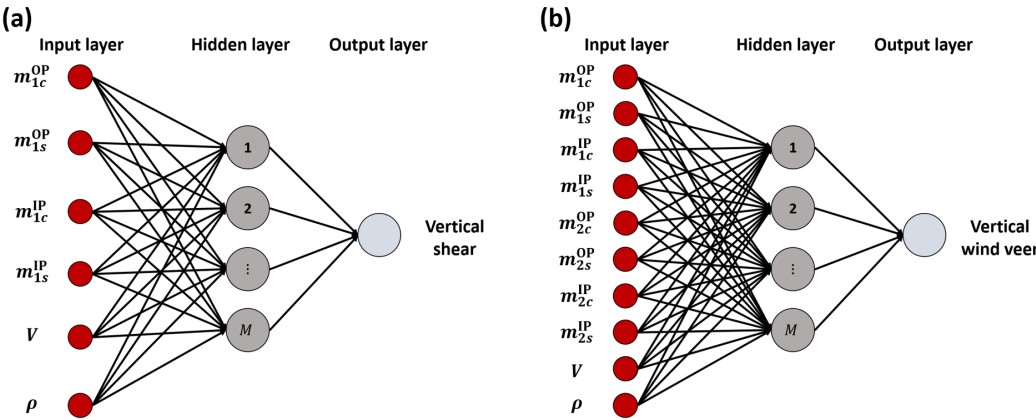

**Figure 1.** Graphical representation of the neural observer of vertical wind shear **(a)** and vertical wind veer **(b)**, with their respective inputs.

## 95   2.2   Test site

The shear and veer observers were identified and validated using wind field and turbine load measurements from the BHV test site (Meyer and Gottschall, 2022), a former airport located in close proximity to Bremerhaven, in the northwest of Germany next to the Weser river. The test site is built around the 8 MW research wind turbine AD8-180. Flat and homogeneous terrain conditions are present in the westerly direction, whereas an urban terrain prevails in the easterly direction. Various wind and
turbine-related measurements have already been carried out at this site, as reported in previously published studies (Giyanani et al., 2022; Huhn and Gómez-Mejía, 2022; Meyer and Gottschall, 2022; Hung et al., 2022; Wegner et al., 2022). The test site is shown in Fig. 2, with a view looking east.



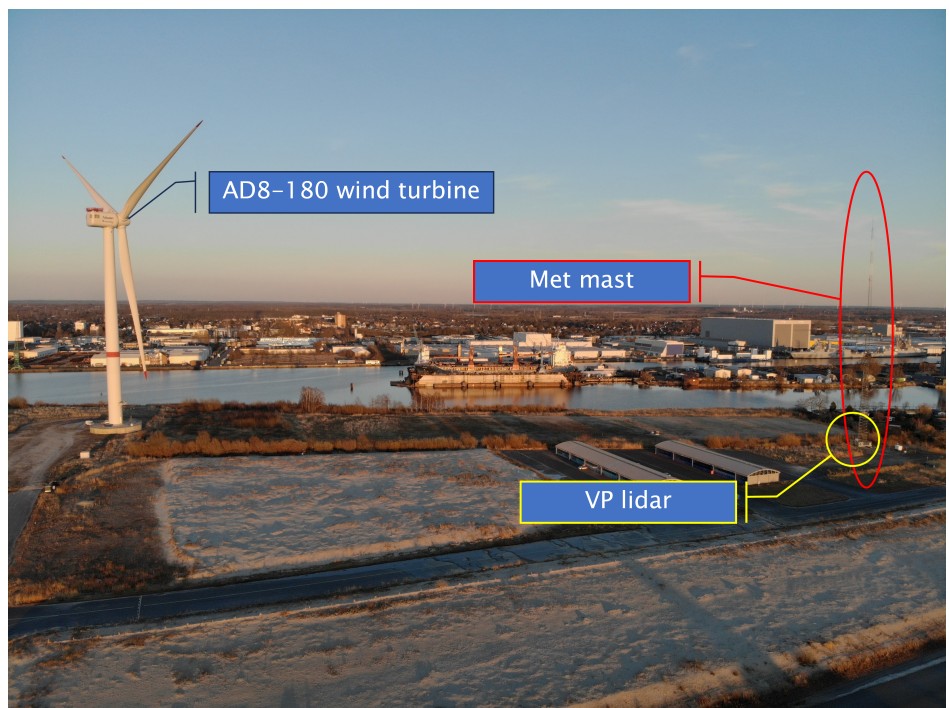

**Figure 2.** View from west to east of the BHV test site, with the AD8-180 wind turbine on the left, and the met mast and VP lidar on the right.

The AD8-180, an 8 MW machine with a 180 m rotor diameter (D) and a 115 m hub height, is equipped with several sensors, including strain gauges placed at various spanwise positions along the blades. Operational data from the SCADA system, together with the strain gauge measurements, are available at a 25 Hz frequency. Flapwise and edgewise measurements from the strain gauges placed at blade root were converted into out and in-plane components, based on blade pitch angle. Next, using the azimuthal rotor position, the load signals were converted into 1 and 2P harmonics, to be used as network inputs (see Eq. 4).

An IEC-compliant met mast is installed at a distance of 399.3 m ($\approx$ 2.2D) from the turbine, in the 189° direction. The mast is equipped with cup anemometers at five heights up to 114.7 m, as well as wind vanes at three heights reaching up to 110 m, i.e. just below the hub. Data from the mast is available at a sampling rate of 1 Hz. Additionally, a barometer, thermometer, and hygrometer are available to derive air density.

A VP lidar of the type WindCube V2 is installed next to the met mast, measuring wind speed and direction at heights from 40 m up to 290 m. Various studies have shown a good agreement between cup anemometers and VP lidars for the measurement of wind speed and direction (Gottschall et al., 2012; Clifton et al., 2018). Furthermore, the VP lidar is an established measurement device for power performance testing and wind resource assessment according to IEC 61400-50-2:2022 (IEC, 2022). The lidar sequentially measures line-of-sight velocities for a fixed scan pattern of 4 beams along a cone with half-opening angle of 28°, combined with one vertically scanning beam. Wind speed and direction are then reconstructed at each measured height from the line-of-sight velocities with every updated line-of-sight measurement, i.e. every 0.8 seconds. As the met mast



provides wind speed and direction measurements only for the lower half of the rotor, the VP lidar is used for measuring these
quantities from a height of 40 m to the top of the rotor.

A sketch of the relevant heights and distances of turbine, met mast and lidar are shown in Fig. 3. This study is based on a data
set of synchronized turbine, mast and lidar measurements collected for 115 days within the period from 30 July to 12 December
2021. The data set contains significant variability of the ambient conditions as well as a large occurrence of southerly winds,
where the met mast is directly upwind of the turbine.

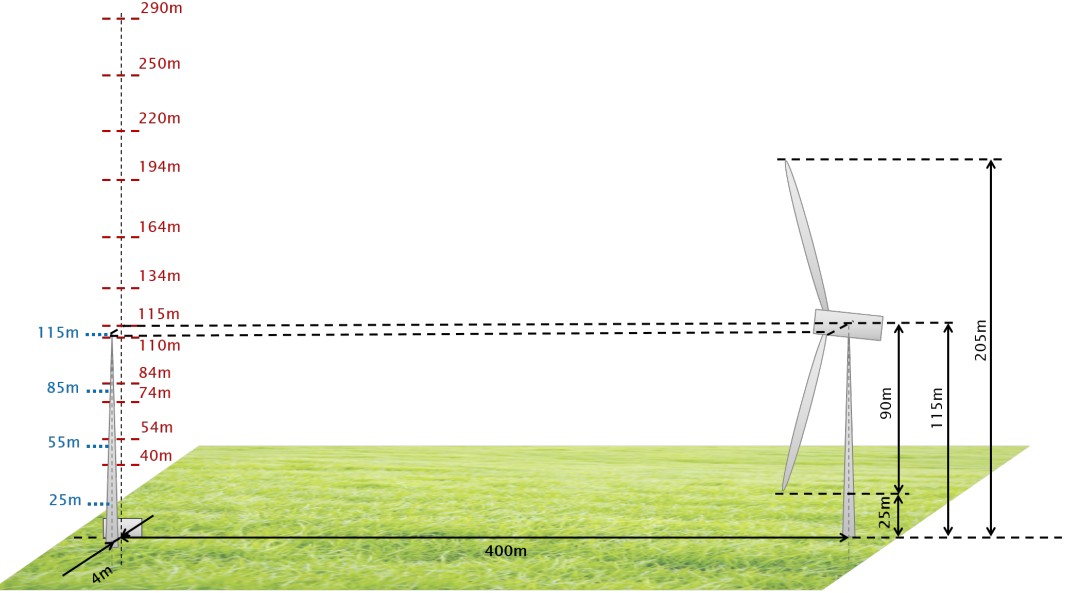

**Figure 3.** Sketch (to scale) of the test site with the relevant dimensions rounded to the next integer. Heights are given relative to ground level
at the turbine location.

## 2.3    Field measurements of shear and veer

The wind observer networks are trained based on measurements of the wind shear and veer, together with their corresponding
network inputs, as expressed in the first term of the objective function given by Eq. (3).

In this work, wind shear and veer measurements were provided by the VP lidar. In fact, the VP lidar measures at 12 heights
from 40 to 290 m above ground, thus including most of the rotor-swept area, which ranges from the lower blade tip (LBT)
point $z_{\mathrm{LBT}} = 25$ m to the higher blade tip (HBT) point $z_{\mathrm{HBT}} = 205$ m.

Both for shear and veer, a linear best fit was first computed using the nine VP lidar measurements of wind speed $V$ and
direction $\Gamma$ included within the rotor-swept area, i.e. between 40 and 195 m above ground (see Fig. 3). Next, shear and veer
were computed as

$$\kappa_v = \frac{V(z_{\mathrm{HBT}}) - V(z_{\mathrm{LBT}})}{z_{\mathrm{HBT}} - z_{\mathrm{LBT}}}, \tag{7a}$$





$$\Delta\theta = \frac{\Gamma(z_{\mathrm{HBT}}) - \Gamma(z_{\mathrm{LBT}})}{z_{\mathrm{HBT}} - z_{\mathrm{LBT}}}, \tag{7b}$$

where the terms at the numerators of these two expressions are computed via the linear fit evaluated at the lower and higher blade tip points, respectively.

Before using these lidar-based rotor-effective wind characteristics for training, their accuracy was verified against the IEC-compliant met mast present at the site. Vertical shear and veer were derived from the mast measurements following the same linear-fitting procedure previously described for the lidar, i.e. using

$$\kappa_{v\,\mathrm{Low}} = \frac{V(z_{\mathrm{HUB}}) - V(z_{\mathrm{LBT}})}{z_{\mathrm{HUB}} - z_{\mathrm{LBT}}}, \tag{8a}$$

$$\Delta\theta_{\mathrm{Low}} = \frac{\Gamma(z_{\mathrm{HUB}}) - \Gamma(z_{\mathrm{LBT}})}{z_{\mathrm{HUB}} - z_{\mathrm{LBT}}}. \tag{8b}$$

Notice that, since the mast reaches only up to hub height $z_{\mathrm{HUB}}$, the resulting shear and veer are defined over only the lower half of the rotor disk.

To perform a valid comparison, also the shear and veer derived from the lidar were computed over the lower part of the rotor, which was obtained by considering only measurements in the range from 40 to 115 m. Figure 4 shows the results of this comparison in the form of 10 min averages, reporting the met mast measurements on the $x$ axis and the corresponding lidar quantities on the $y$ axis. Wind speeds at hub height, shown in Fig. 4a, have a high Pearson coefficient $R$ of 0.99 and a mean absolute error (MAE) of about 1.22 ms$^{-1}$. There is a high correlation also for wind shear and veer, which have Pearson coefficients of 0.97 and 0.95, respectively, as shown in Fig. 4b and c. In addition to the different measurement technology, differences might be caused by the fact that the mast reaches down to 25 m above the terrain, whereas the lowest measurement point for the lidar is at 40 m. Figure 4c suggests the existence of a slight slope difference for veer. This might be caused by the met mast, because vertical veer is obtained only from two heights above ground, and possibly because of some minor misalignment of its wind direction sensors. Given its uncertain origin, lidar measurements were not recalibrated to eliminate this effect.

## 3 Results

The wind observers for shear and veer formulated in Sect. 2.1 were tested on a dataset collected at the site described in Sect. 2.2. The next pages explain first in Sect. 3.1 the identification of the observers from a subset of the data. Next, Sect. 3.2 presents the results obtained by using the observers on an independent validation subset.

### 3.1 Observer identification

The data set was cleaned, to retain only data points when the turbine was operational and all necessary measurements (SCADA, strain gauges, lidar) were available. This resulted in about 18 full days of valid data points, of which about 67% (i.e. about 290 hours) were used for training, leaving the rest (i.e. about 138 hours) for validation. The ranges and number of occurrences of values of wind speed $V$, air density $\rho$, vertical shear $\kappa_v$, and veer $\Delta\theta$ in the two data sets are shown in Fig. 5.

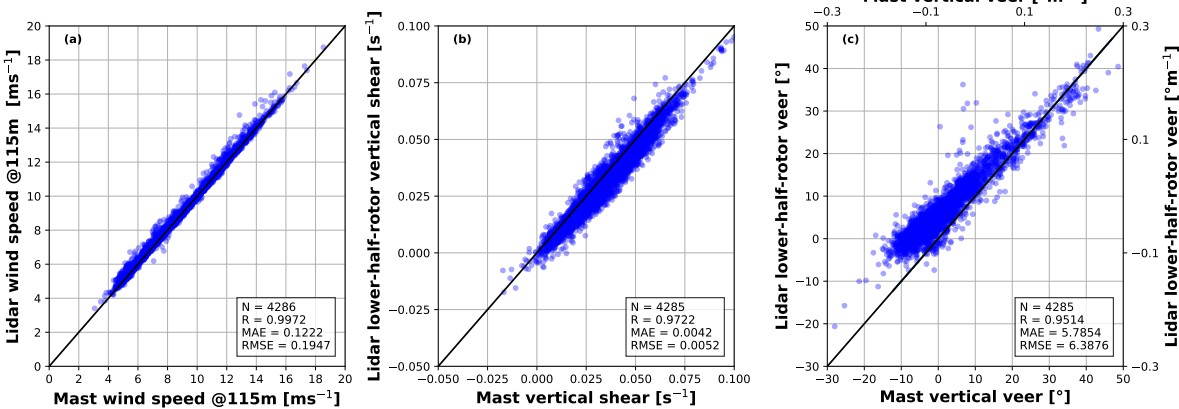

**Figure 4.** Correlation of measured quantities from the met mast ($x$ axis) and VP lidar ($y$ axis). Hub-height wind speed **(a)**; lower-half-rotor vertical shear **(b)**; lower-half-rotor veer, both in absolute [°] (i.e. between $z_{\mathrm{LBT}}$ and $z_{\mathrm{HUB}}$) and in relative [°m$^{-1}$] terms **(c)**. Black solid line: ideal match; R: Pearson's correlation coefficient; N: number of data points; MAE: mean absolute error; RMSE: root mean square error.

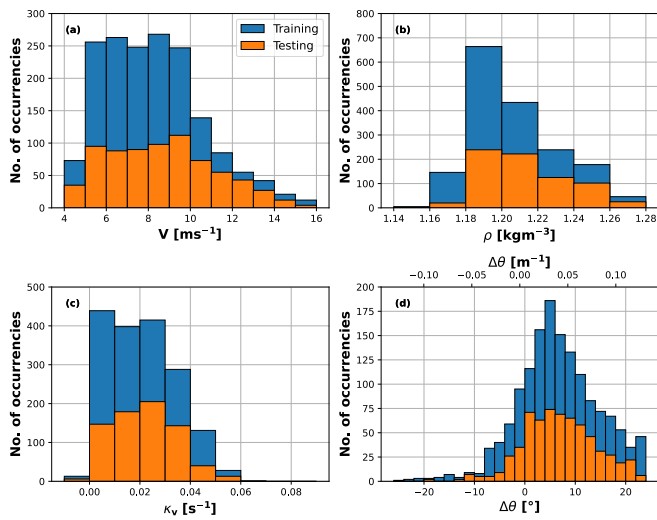

**Figure 5.** Range and number of occurrences of 10 min averages of wind speed **(a)**, density **(b)**, vertical shear **(c)**, and wind veer **(d)**. Blue: training data set; red: validation data set.





Air density $\rho$ and wind speed $V$ appearing in the network inputs (see Eq. 4) were measured as follows. Air density was derived from the available measurements of pressure, temperature, and humidity, using the ideal gas laws. Wind speed was obtained by means of an observer based on the standard SCADA signals of power, rotor speed and blade pitch (Soltani et al., 2013). The use of an observer, which is based only on standard operational data, renders the shear and veer observers usable on common production machines, where a lidar or a neighboring met mast might not be available. A MoWiT model (Fricke

et al., 2021) of the AD8 turbine was used to generate offline a look-up table (LUT), storing the dependency of produced power on ambient wind speed, pitch angle and rotor speed, considering mechanical losses in the drive-train and the efficiency of the generator. Next, the LUT was inverted by a Newton iteration using the aerodynamic torque obtained from the dynamic torque-balance equation, and on measured power, pitch and rotor speed from the SCADA data stream. Figure 6 shows the correlation between 10 min averages of the rotor-effective wind speed (REWS) from the observer, reported on the $y$ axis, and

the wind speed obtained by averaging the available lidar measurements along the rotor height, reported on the $x$ axis. The match between these two quantities is good, with a Pearson coefficient of 0.98 and a MAE of about 0.35 ms$^{-1}$.

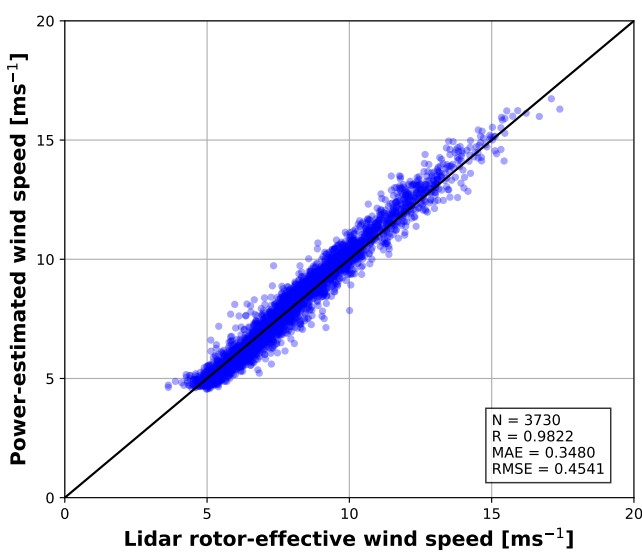

**Figure 6.** Correlation of 10 min averages between the rotor-average wind speed measured by the lidar ($x$ axis) and the rotor-effective wind speed measured by the observer ($y$ axis). Black dashed line: ideal match.

  Both for the shear and veer observers, the best-performing network configuration was found by trial and error to comprise of one single hidden layer and 10 neurons. Both networks took of the order of a few seconds for training on a standard desktop computer.

**3.2 Shear and veer observer performance**

After training, the two observers of shear and veer were tested on the 138-hour-long validation data set.





Figure 7 reports the results in terms of 10 min averages. Quantities estimated by the observers are reported on the $y$ axis, while the lidar-measured references are on the $x$ axis. Figure 7a indicates an excellent match for shear, with a Pearson coefficient $R = 0.947$ and an RMSE of about $4.015 \cdot 10^{-3}$ s$^{-1}$. Figure 7b indicates a slightly lower quality of the results for veer, with $R = 0.879$ and a larger scatter, as quantified by an RMSE of about 5.78°. MAEs are $3 \cdot 10^{-3}$ s$^{-1}$ and 4°, respectively, for shear and veer.

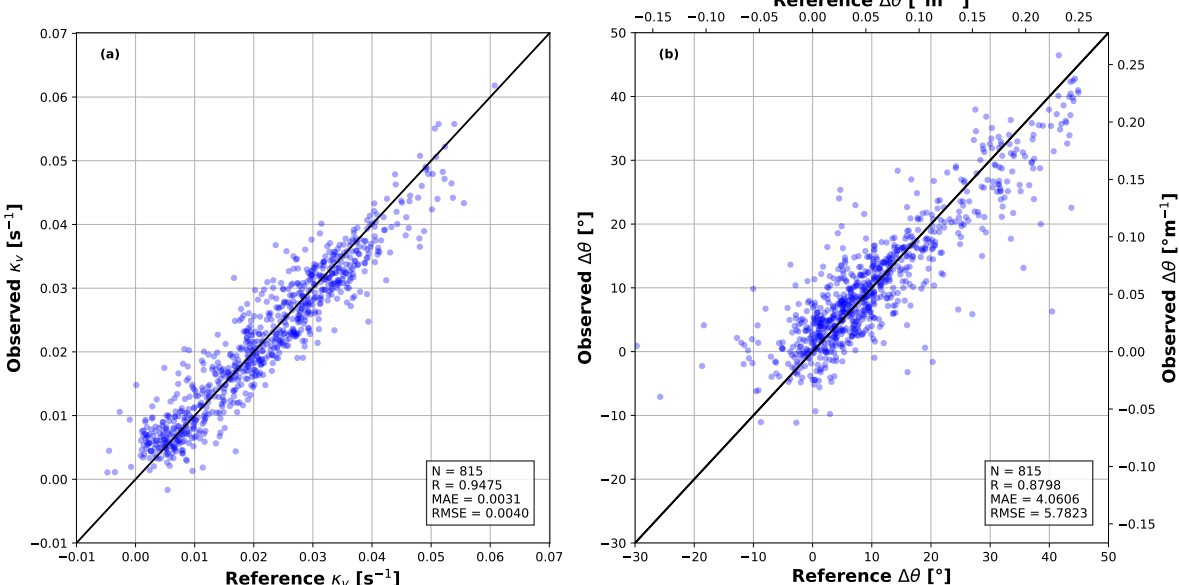

**Figure 7.** Correlation of 10 min averages between estimated wind characteristics ($y$ axis) and their reference lidar-measured quantities ($x$ axis). Vertical shear $\kappa_v$ **(a)**; wind veer $\Delta\theta$ **(a)**. Black dashed line: ideal match.

Exemplary time histories of observed and reference quantities are given in Fig. 8. Figure 8a and b show the estimated (red) and reference (blue) wind shear and veer, respectively. Additionally, Fig. 8c reports the wind direction at the site, where a horizontal black solid line indicates the 189° direction for which lidar and turbine are aligned. The observation of shear and veer was performed at 1 Hz, and results were then averaged with a 1 min moving window. The figures indicate that the observed quantities follow quite well their respective references, both in terms of trends and mean values. There is a particularly good match between 5 am and 7 am, when lidar and turbine are aligned, although some of the worst match is between 4 am and 5 am, when the two are also almost aligned. However, the two observes are clearly capable of detecting the diurnal cycle, characterized by a higher shear and veer during the night, but also rapid events such as the spike observed around 9:30 am. The quality of these results seems to be more than capable of supporting applications such as the one described in Sucameli et al. (2023), where one needs to account for the effects of shear and veer on wake development for an effective implementation of wind farm control.



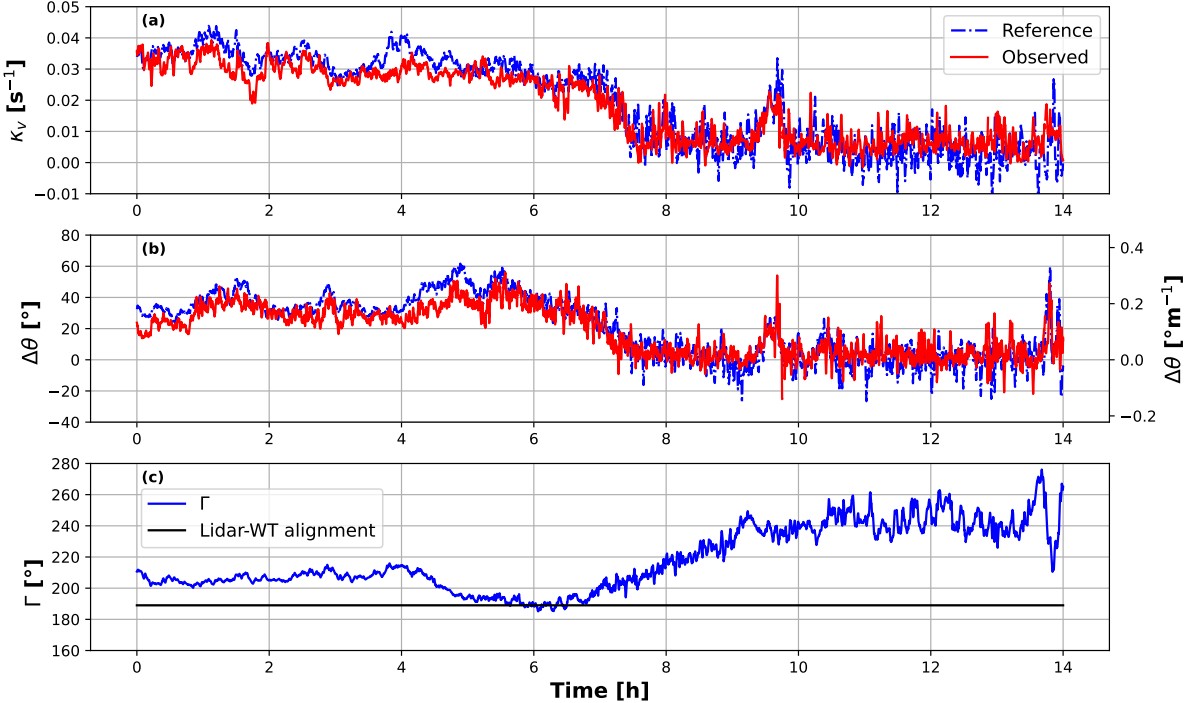

**Figure 8.** Time histories of vertical shear **(a)**, wind veer **(b)**, and wind direction measured at the mast **(c)**. Blue: observed quantities; red: lidar-measured reference.

Finally, to provide with more statistically relevant results, Fig. 9a and b respectively show the MAEs of the observed shear and veer as functions of wind speed, for different turbulence intensity (TI) values. Figure 9c reports the number of available data hours for each specific speed and TI bin. MAEs were computed after averaging 1 Hz observations over 10 minutes, and then comparing them with their respective lidar-measured references. Interestingly, both the observed shear and veer appear to be rather insensitive to TI and wind speed, only the shear error exhibiting a growing trend for low wind speeds. These findings should however be confirmed with a larger data set, since the different bins used in this analysis are not equally populated.

## 4 Conclusions

This paper has demonstrated that it is possible to observe vertical wind veer from the operational response of a large wind turbine. The paper also performed the first validation of the observation of shear and veer over the full rotor height with respect to reference measurements obtained with a VP lidar. The study was conducted at the BHV test site using the highly instrumented 8 MW AD8-180 wind turbine. Additionally, the presence on site of an IEC-compliant met mast allowed for a comparison – although limited only to the lower half of the rotor – of the lidar-measured wind speed, shear and veer, enhancing the confidence in the results.



**WIND
ENERGY
SCIENCE**
DISCUSSIONS

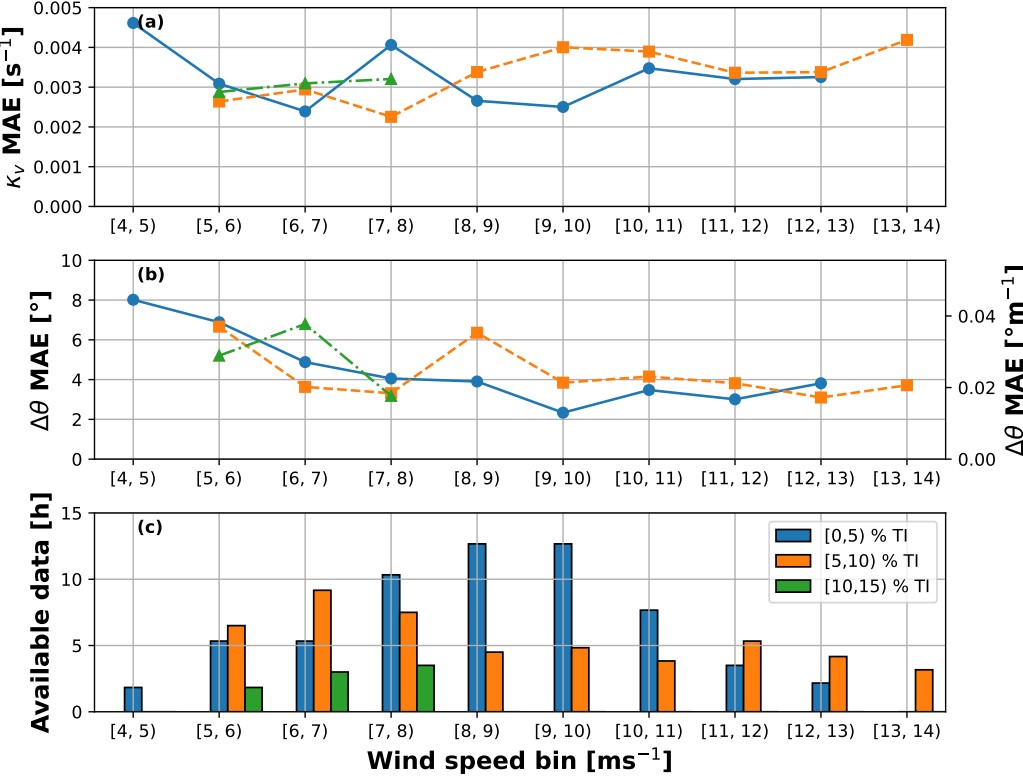

**Figure 9.** MAE of vertical shear $\kappa_v$ **(a)** and wind veer $\Delta\theta$ **(b)**, and hours of available data **(c)** vs. binned rotor-effective wind speed $V$, for different TI levels.

Based on the results reported herein, the following conclusions can be drawn:

- The correlation between 10 min averages of the observed and lidar-measured shear and veer results in Pearson coefficients of $R = 0.947$ and $R = 0.879$, respectively. The quality of shear is presumably better because its observation relies only on 1P harmonics, whereas veer requires also the 2P components, which are probably more affected by turbulence.


- For the same reason, veer has a higher scatter than shear, as seen in Fig. 7.

- Both the shear and veer observers seem capable of tracking both slow and relatively fast changes in ambient conditions. In particular, the exemplary time history reported in Fig. 8 indicates the ability to follow rapid changes of the duration of the order of tens of minutes with good accuracy. The examination of other similar time histories, not reported here for brevity, supports and confirms this finding.


- Results aggregated over the whole validation data set indicate typical MAEs for veer in the neighborhood of 4° and for shear around $3 \cdot 10^{-3}$ s$^{-1}$. For both quantities, there is apparently a good robustness with respect to TI, which has only





minor effects on the quality of the results. Additionally, there is also a general insensitivity to the operating wind speed, some degradation appearing only for veer at the lowest wind speeds.

In general, these results seem to indicate the ability of the harmonic-based observers to estimate shear and veer from the operational response of a wind turbine, with a close match to the widely adopted vertical profiling lidar. In evaluating these results, however, two remarks are in order:

- The lidar measurements cannot be assumed as an absolute ground truth. In fact, as all measurements, they are affected by various sources of error, and they represent spatial and temporal averages that differ from the ones performed by the observer and by the anemometry installed on the mast. Additionally, the mast is not exactly co-located with the turbine,
it is not always exactly in front of it, and it is does not even span exactly the same height as it starts measuring a bit above the LBT. Therefore, an exact match between observers and lidars cannot and should not, in general, be expected. One source of discrepancy could be removed by the use of a forward staring lidar, which would at least always provide wind measurements directly upwind of the turbine.

- Some of the speed and TI bins are not well populated, which might have some effect on the significance of the perfor-
mance statistics. This source of uncertainty could be removed by the use of longer data sets, which however were not available for this study.

## Appendix A: Harmonic content of the observers

Following Eggleston and Stoddard (1987), the flow speed components normal (noted $u_n$) and tangential (noted $u_t$) to the rotor disk can be written

$$u_n = V(1-a) - V\kappa_s \frac{r}{R}\cos\psi - V_0\beta\sin\psi, \tag{A1a}$$

$$u_t = \Omega r, \tag{A1b}$$

where $a$ is the axial induction, $\beta$ is the blade flap angle, $\psi$ is the azimuthal blade position (where $\psi = 0$ when the blade is vertical pointing downwards), $r$ is a spanwise position, $R$ the rotor radius, $V_0$ is the cross-flow (i.e., a lateral wind speed component parallel to the rotor disk), and $\Omega$ the rotor speed. When the inflow is veered, the cross-flow can be written as

$V_0 = \Delta\theta(r/R)\cos\psi$. The flapwise bending moment on the blade is obtained by integrating the lift $L$ along the blade span, where

$$L = \frac{1}{2}\rho u^2 c C_{L_\alpha}\alpha \approx \frac{1}{2}\rho c(u_p u_t - \theta u_t^2), \tag{A2}$$

where $u \approx u_t$ is the flow speed at the blade section, $c$ is the chord, $C_{L_\alpha}$ is the lift slope, $\alpha \approx u_p/u_t - \theta$ the angle of attack (considering small angles), and finally $\theta$ is the pitch angle.

Inserting the second expression of Eq. (A2) into Eq. (A1), it follows that lift, and hence bending moments, depends on terms proportional to $\kappa_s\cos\psi$ and $\Delta\theta\sin\psi\cos\psi = 2\Delta\theta\sin(2\psi)$. Hence, shear leaves a mark on the 1P harmonic of blade loads, and veer on their 2P harmonics.



*Data availability.* Data from the field measurements can be requested to JG. All figures and the data used to generate them can be retrieved in Pickle Python and MATLAB formats via https://doi.org/10.5281/zenodo.8335021 (Bertelè et al., 2023).

*Author contributions.* CLB developed the concept of the wind rotor as a sensor, formulated the harmonic-based neural observers in collaboration with MB, and co-supervised the research together with JG. MB implemented the observers, performed all numerical observations, their pre- and post-processing and analysis. CRS conceptually contributed to the pre-processing analysis. PJM and MB processed the raw data from the field measurements. JF performed the aeroservoelastic simulations for the wind speed observer, MB post-processed their results. All authors equally contributed to the interpretation of the results. MB and CLB wrote the manuscript, with contributions by CRS and by JG,
PJM, JF, and AW in the description of the test site and the processing of the lidar data. All authors provided important input to this research work through discussions and feedback and by improving the manuscript.

*Competing interests.* The contact author has declared that none of the authors has any competing interests, except for CLB who is the Editor in Chief of the Wind Energy Science journal.

*Financial support.* This work is supported in part by the Power Tracker (FKZ: 03EE2036A), Life Odometer (FKZ: 03EE3037B), HighRE
(FKZ 03EE2001) and Testfeld BHV (FKZ 0324148) projects, which receive funding from the German Federal Ministry for Economic Affairs and Climate Action (BMWK). This work has also been partially supported by the MERIDIONAL project, which receives funding from the European Union's Horizon Europe Programme under the grant agreement No. 101084216.



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
