# Peer review of "The rotor as a sensor — Observing shear and veer from the operational data of a large wind turbine"

_Wind Energy Science, 2023_

## Referee Comment (RC1)

**Review - The rotor as a sensor - Observing shear and veer from the operational data of a large wind turbine**

December 4, 2023

**Summary**

This is a good paper on estimation of shear and veer using NN. It is tested using a full scale turbine. This is a contribution on its own. Also the performance of the estimators are convincing. It would be interesting to see the NN estimators compared with something simpler. The paper is well written and organized. Consequently, the paper can be published after minor corrections.

**Specific comments**

1. P5 "Flapwise and edgewise measurements from the strain gauges placed at blade root"

    (a) Is the sensors used standard for the wind turbine?

    (b) Or is it part of the measurement campaign sensors?

    (c) It would be best if it were the standard ones. This is also what is stated by the paper on p 9 l 160-170.

2. P13 Appendix A There are some unclear parts here

    (a) If $u_n$ is tangential to the rotor disc and $V_0$ is parallel to the rotor disc, how can $V_0$ contribute to $u_n$?

    (b) "$\beta$ is the blade flap angle" but what is that?

    (c) What is $u_p$? Is it $u_n$

---

## Author Comment (AC1)

**Reply to Reviewers**

The authors would like to thank the two reviewers for their time and for the useful feedback. All inputs that they provided have contributed to the improvement of the paper.

A list of point-by-point replies to the reviewers' comments is reported in the following.

We have taken this opportunity to make several small editorial changes to the text, in order to improve readability. A revised version of the manuscript is attached to the present reply, with additions highlighted in blue and deletions marked in red.

The authors

**Reviewer 1**

*Summary: This is a good paper on estimation of shear and veer using NN. It is tested using a full scale turbine. This is a contribution on its own. Also the performance of the estimators are convincing. It would be interesting to see the NN estimators compared with something simpler. The paper is well written and organized. Consequently, the paper can be published after minor corrections.*

*Specific comments:*

1. **Reviewer**: *P5 "Flapwise and edgewise measurements from the strain gauges placed at blade root"*
   *(a) Is the sensors used standard for the wind turbine?*
   *(b) Or is it part of the measurement campaign sensors?*
   *(c) It would be best if it were the standard ones. This is also what is stated by the paper on p 9 l 160-170. 2.*
   **Authors**: Strain gauges installed on turbine blades are becoming part of the standard instrumentation on newer machines, since they are nowadays used for load-control and condition monitoring purposes. However, in the specific case of this AD8, the turbine was equipped with optical strain gauges and additional instrumentation in the framework of an unrelated research project. This has now been noted in the manuscript.

2. **Reviewer**: *P13 Appendix A There are some unclear parts here:*
   *(a) If un is tangential to the rotor disc and V0 is parallel to the rotor disc, how can V0 contribute to $u_n$?*
   *(b) "$\beta$ is the blade ap angle" but what is that?*
   *(c) What is $u_p$? Is it $u_n$*
   **Authors**: Thank you for your comments, indeed this part was not clearly written and contained various imprecisions. We have now corrected and expanded the description, and we have also added a figure to clarify the meaning of the various symbols.

**Reviewer 2**

*This is an interesting, well-written, concise paper that presents field validation of neural network-based wind shear and veer observers using blade root bending moment measurements. The study builds on previous work by some of the authors that discussed a neural network-based observer for vertical shear and yaw misalignment. The current study is a valuable contribution to the literature because it presents the first field validation of a veer observer known to the authors and validates the shear and veer observers using reference wind measurements across the full height of the rotor using a ground-based vertically profiling lidar.*

*Although I don't have any major concerns about the paper, there are many places where additional details or clarification should be provided, as explained in the comments below.*

*Specific comments:*

1. **Reviewer:** *Section 2.1: A few more details about the neural network-based shear and veer observers would be helpful for readers, especially non-experts. In particular, after equation 1, can you explain what "p" represents? It is explained a couple paragraphs later, but it would be better to mention the variable as soon as it is shown in Eq. 1. Some discussion or references about "sigmoid activation functions" and the "backpropagation" training method should be provided for those unfamiliar with neural networks. Lastly, how is the number of hidden neurons M hyperparameter chosen? Is this optimized as part of the observer identification (could also be discussed in Section 3.1)?*
   **Authors**: We have anticipated the definition of p, as suggested. Three references (Bishop, 2006; Burden and Winkler, 2009; Matlab 2023) contain ample material and all necessary information for non-experts. The choice of M is indeed explained at the very end of Sect. 3.1.

2. **Reviewer:** *Ln. 91: "30 sec moving average of the rotor-effective wind speed": I suggest adding "estimated" rotor-effective wind speed.*
   **Authors**: Thank you, we implemented your suggestion.

3. **Reviewer:** *Figure 3: I suggest labeling the lidar measurement heights in the figure as belonging to the "lidar", or mentioning this in the caption, to make it clear what the heights refer to.*
   **Authors**: Thank you, we implemented the suggestion by adding in the caption the color-coding for the labels of the measurement heights.

4. **Reviewer:** *Section 2.3: Why don't you use the more common power law shear exponent definition of vertical wind shear in the estimator instead of the linear shear? This would likely be a more useful term for many applications of the observer. I understand that the blade load harmonics are more easily connected to linear shear across the rotor, but since a neural network is used, it seems the NN could relatively easily be trained to estimate the equivalent power law shear as well.*
   **Authors**: This is indeed possible, but theoretically it would require higher harmonics. The reason for this is explained in Kim et al., 2022. We have expanded the text in Sect. 2.1 in order to clarify this point.

5. **Reviewer:** *Ln. 149: Fig. 4a lists 0.122 as the MAE, but the text states 1.22 m/s. Is there a typo in the text?*
   **Authors**: Yes, thank you. We have corrected the typo.

6. **Reviewer:** *Section 3.1: Please clarify what data sample period is used for the observer identification. In Section 3.2, it appears that the observers use 1 Hz input data. Is this the same frequency used for training the observer? Do you expect the trained observers can be*

*applied to data with a range of sampling frequencies, or should they only be used with data of the same frequency as the training data?*
**Authors**: We have modified the text of Sect. 3.1 and 3.2 to better address this topic.

7. **Reviewer:** *Ln. 162: "of which about 67%... were used for training": Can you explain how the training data points were selected? Are they randomly distributed throughout the data set, or are the first 67% of the data used for training and the last 33% used for validation?*
**Authors**: The dataset for training was selected at random. This was done to account for the fact that, although the dataset comprises about 430 hours, these hours are spread over several months (from August to December) that exhibit significant seasonal variabilities. The text was updated to improve clarity on this point.

8. **Reviewer:** *Fig. 4 caption: Please mention that 10-minute averages are shown in this figure.*
**Authors**: We have adapted the caption.

9. **Reviewer:** *Ln. 172: "using the aerodynamic torque obtained from the dynamic torque-balance equation, and on measured power, pitch and rotor speed from the SCADA data stream." The dynamic torque-balance equation generally requires the measured generator torque and generator acceleration. Should these measurements be added to the list here?*
**Authors**: Torque is measured on this machine, but the data appeared to be not sufficiently reliable. Therefore, the implementation was based on power. This has the additional advantage that power measurements are typically available in the SCADA data stream. As explained in the text, power is converted into torque by accounting for electrical and mechanical losses, and by dividing by the measured rotor speed. The text was updated to explain that acceleration is obtained by deriving the measured rotor speed, as suggested.

10. **Reviewer:** *Ln. 194: "The quality of these results seems to be more than capable of supporting applications such as the one described in Sucameli et al.": I don't think this is incorrect necessarily, but it's kind of an arbitrary statement. Can you explain this a little more? What kind of accuracy/quality are you looking for in the estimates for them to be considered capable?*
**Authors**: We believe that the meaning of our statement was quite clear. However, it is also clear that the only way to prove the statement would be to actually use these estimated quantities to support the implementation of a wind farm controller. As this has not been done within the scope of the present research, we have now eliminated this sentence from the manuscript.
Additionally, since the cited paper Sucameli et al., 2023 has not been submitted yet, we have completely removed it from the list of references. In the absence of this paper, in the introduction the motivation for the need of shear and veer measurements has now been completely rewritten, including the addition of three new references.

11. **Reviewer:** *Ln. 201: "Interestingly, both the observed shear and veer appear to be rather insensitive to TI and wind speed, only the shear error exhibiting a growing trend for low wind speeds.": This statement doesn't quite seem correct. At most higher wind speeds (>= 8 m/s), there is clearly higher error for higher turbulence levels in Fig. 9. Further, both shear and veer errors appear to exhibit a growing trend at low wind speeds.*
**Authors**: We agree and we have reworded the text with more cautious comments, especially at the light of the scarce population of some wind speed bins.

12. **Reviewer:** *Ln. 217: "the ability to follow rapid changes of the duration of the order of tens of minutes": Should this be tens of seconds? Depending on the application, it doesn't seem correct to consider tens of minutes as rapid. Or can you discuss this further?*
**Authors**: Thank you, we have updated the text as suggested.

13. **Reviewer:** *Ln. 221: "there is a good robustness with respect to TI, which has only minor effects on the quality of the results.": As mentioned in comment 11, there is significantly larger error for higher TI at some higher wind speeds.*
    **Authors**: We have rephrased the text.

14. **Reviewer:** *Ln. 223: "some degradation appearing only for veer at the lowest wind speeds.": As also mentioned in comment 11, based on Fig. 9 it seems this is the case for shear as well.*
    **Authors**: We have rephrased the text.

15. **Reviewer:** *Conclusion section: Can you comment on whether you expect the trained observer can be applied to other turbines of the same model? Or would the training procedure need to be applied to each individual turbine?*
    **Authors**: Thank you for this comment. Unfortunately we have no direct evidence yet that this is possible. However, we have used this argument (as we already did in previous publications) to further justify the use of a very limited harmonic content in the observer. This discussion has now been added to Sect. 2.1.

16. **Reviewer:** *Ln. 242: How is the blade flap angle beta defined? Further, what is the sign convention of the azimuthal blade position and cross-flow velocity? A figure would be helpful here.*
    **Authors**: We have expanded this section, corrected some imprecisions and added a figure.

17. **Reviewer:** *Eq. A2: Can you provide a reference for this equation? Also, what is "u_p"? Should this be "u_n"? Lastly, it would be good to clearly define rho as the air density as well.*
    **Authors**: This part has been rewritten, and hopefully it is now clearer. Air density is not defined here, because it is has already been introduced in Sect. 2.1.

18. **Reviewer:** *Ln. 250: "Inserting the second expression of Eq. (A2) into Eq. (A1)…": It isn't clear where this would be inserted in Eq. A1. Please state where lift appears in Eq. A1. Or do you mean inserting the second expression of A1a into A2?*
    **Authors**: Thank you, this has now been corrected.

19. **Reviewer:** *Ln. 251: "shear leaves a mark on the 1P harmonics of blade loads, and veer on their 2P harmonics". Please describe why the veer observer also uses the 1P harmonics as inputs, if veer appears to only depend on 2P harmonics as shown here.*
    **Authors**: We tested both an implementation based only on 2P harmonics, and one based on 1 and 2P harmonics. The latter appeared to provide slightly better results, and was therefore used for producing the results of the paper. The reason for this is presumably due to the approximate nature of the model that we used to estimate the harmonic content of the observer, and that includes several assumptions and simplifications. This discussion has now been added to the revised manuscript.

[revised manuscript text omitted]